# Combining radio-telemetry and radar measurements to test optimal foraging in an aerial insectivore bird

Itai Bloch[1]*, David Troupin[1], Sivan Toledo[2], Ran Nathan[3], Nir Sapir[1]

[1]Department of Evolutionary and Environmental Biology and Institute of Evolution, University of Haifa, Haifa, Israel; [2]Blavatnik School of Computer Science, Tel-Aviv University, Tel Aviv, Israel; [3]Department of Ecology, Evolution, and Behaviour, Alexander Silberman Institute of Life Sciences, The Hebrew University of Jerusalem, Jerusalem, Israel

## eLife Assessment

This **important** study enhances our understanding of the foraging behavior of aerial insectivorous birds. Using **solid** methodology, the authors have collected extensive data on bird movements and prey availability, which in turn provide support for the main claim of the study. The work will be of broad interest to behavioral ecologists.

**\*For correspondence:**
itaibloch2@gmail.com

**Competing interest:** The authors declare that no competing interests exist.

**Abstract** Optimal foraging theory posits that foragers adjust their movements based on prey abundance to optimize food intake. While extensively studied in terrestrial and marine environments, aerial foraging has remained relatively unexplored due to technological limitations. This study, uniquely combining BirdScan-MR1 radar and the Advanced Tracking and Localization of Animals in Real-Life Systems biotelemetry system, investigates the foraging dynamics of Little Swifts (*Apus affinis*) in response to insect movements over Israel's Hula Valley. Insect movement traffic rate (MoTR) substantially varied across days, strongly influencing swift movement. On days with high MoTR, swifts exhibited reduced flight distance, increased colony visit rate, and earlier arrivals at the breeding colony, reflecting a dynamic response to prey availability. However, no significant effects were observed in total foraging duration, flight speed, or daily route length. Notably, as insect abundance increased, inter-individual distances decreased. These findings suggest that Little Swifts optimize their foraging behavior in relation to aerial insect abundance, likely influencing reproductive success and population dynamics. The integration of radar technology and biotelemetry systems provides a unique perspective on the interactions between aerial insectivores and their prey, contributing to a comprehensive understanding of optimal foraging strategies in diverse environments.

## Introduction

Optimal foraging theory predicts how foragers should adjust their movement and behavior based on the costs and benefits of finding and consuming food (*Pyke et al., 1977*; *Pyke, 1984*; *Schoener, 1971*; *Emlen, 1966*; *MacArthur and Pianka, 1966*). Empirical studies have tested optimal foraging predictions in terrestrial and marine environments (*Kramer and Nowell, 1980*; *Crowder, 1985*; *Lincoln and Quinn, 2019*; *Staniland et al., 2010*; *Holder and Polis, 1987*), yet, to the best of our knowledge, no study has thus far utilized advanced tracking tools to empirically examine optimal foraging predictions of foragers in the highly dynamic aerial habitat (*Diehl, 2013*). Understanding optimal foraging in aerial habitats is essential for comprehending complex interactions and adaptations in this dynamic

environment. We combine aerial insect abundance data collected using the BirdScan-MR1 radar (*Zaugg et al., 2008*; *Liechti et al., 2019*; *Nilsson et al., 2018*; *Knop et al., 2023*) with measurements of the movement of insectivore birds using the automated and accurate Advanced Tracking and Localization of Animals in Real-Life Systems (ATLAS) biotelemetry system (*Toledo et al., 2020*). This study examines whether the Little Swift (*Apus affinis*), a monomorphic, small insectivore (12 cm, 25 g) that breeds in small colonies and often forages in groups (*Collins et al., 2010*; *Shirihai et al., 1996*; *Paz, 1987*; *Chantler and Boesman, 2021*), optimizes its foraging behavior in response to variations in insect density in the airspace, within the framework of optimal central-place foraging. We note that in a preliminary study we found no discernible differences in foraging characteristics between males and females (*Bloch et al., 2019*).

Aerial insectivores feed on insects (*Hallmann et al., 2014*; *Møller, 2019*; *Bowler et al., 2019*) that have recently been reported to be in decline in different ecosystems and regions of the world (*Hallmann et al., 2014*; *Møller, 2019*; *Hallmann et al., 2017*; *Wagner, 2020*; *Benton et al., 2002*). Among aerial foragers, swifts are highly adapted to life on the wing due to their high flight capabilities, allowing them to undertake different activities in the air and stay airborne for long periods (*Rattenborg, 2006*; *Sachs, 2017*; *Hedenström et al., 2016*; *Bäckman et al., 2001*; *Henningsson et al., 2010*; *Hedrick et al., 2018*; *Liechti et al., 2013*). Nevertheless, during the breeding season, birds return to their central-place breeding colony and provide food to their young throughout the day. Consequently, they may adjust their foraging in relation to different environmental conditions to maximize the net energy obtained during foraging (*Emlen, 1966*; *Charnov, 1976*; *Pianka, 1976*). According to the theory of central-place foraging, traveling to a distant destination is an expensive investment in terms of time and energy compared to traveling to a nearby destination (*Orians and Pearson, 1979*; *Bell, 1990*). Therefore, animals are expected to prefer reducing the time and distance of travel to the food patch and thus will travel farther only when their prey is not sufficiently available near the central place. We thus hypothesize that, in times of abundant food, birds will optimize energy conservation by foraging closer to the colony (*Orians and Pearson, 1979*; *Olsson et al., 2008*). Consequently, we anticipate a reduction in both the average daily air flight distance (hereafter average distance) (Prediction 1) and the maximum daily air flight distance (hereafter maximum distance) (Prediction 2) from the breeding colony under conditions of increased food abundance. This will also result in shorter overall daily flight route length (hereafter daily route) (Prediction 3) and overall daily foraging flight duration (hereafter foraging duration) (Prediction 4). Since breeding swifts may maximize food provisioning to the young, the visitation rate could also be tailored to the abundance of insects (*Pyke, 1984*; *Schoener, 1971*; *Orians and Pearson, 1979*) such that higher food density will facilitate a higher rate of visits at the nest (*Pyke, 1984*; *Kramer and Nowell, 1980*) (Prediction 5). Furthermore, a bird's flight speed, when feeding its young, is expected to vary with food abundance, and this rarely tested prediction suggests an increase in flight speed with greater food abundance (*Norberg, 1981*; *Hedenstrom and Alerstam, 1995*) (Prediction 6). The timing of morning emergence from the colony and evening return to the colony is affected by a number of factors (*Bednekoff and Houston, 1994*; *Bonter et al., 2013*; *Amichai and Kronfeld-Schor, 2019*; *Salamolard and Weimerskirch, 1993*; *Harding et al., 2007*; *Metcalfe and Ure, 1995*). These include predation risks that vary throughout the daily cycle and the optimization of foraging time in relation to food abundance. We hypothesize that the time of arrival at the colony for the night roost and the time of departure from the colony in the morning will be affected by the abundance of flying insects. We specifically predict that swifts will arrive at the colony earlier for roosting when food abundance is sufficiently high to provide enough food for their own and their young's needs (Prediction 7). If insect abundance is correlated in time such that birds may be able to predict insect abundance based on that of the previous day, we expect the swifts' departure time to be delayed when the abundance of insects on the previous day is higher (Prediction 8) as there is no need to maximize the foraging duration if food is abundant and this could reduce predation risk by avian predators that are active early in the morning (*Lang et al., 2019*; *Roth and Lima, 2007*). Consequently, the predicted swifts' emergence times are expected to correlate with the roosting time from the previous night (Prediction 9a). Yet, if no between-day correlation in insect abundance exists, morning departure timing will not be related to insect abundance of the previous day and the two measures will not be correlated (Prediction 9b). For social foraging animals, local enhancement can provide several advantages, including increased energy intake (*Brown, 1988*; *Flemming et al., 1992*; *Krebs, 1974*), higher fitness (*Giraldeau and Caraco, 2000*), improved food detection (*Cvikel et al.,*

2015; *Bijleveld et al., 2015*), and avoidance of predators (*Giraldeau and Caraco, 2000*; *Sorato et al., 2012*). However, an enlarged group size could exacerbate inter-individual competition and may lead to diminished foraging efficiency (*Giraldeau and Caraco, 2000*; *Beauchamp, 1998*). Conversely, increased food abundance ensures adequate sustenance for more group members, thereby alleviating competition. We posit that higher insect abundance would lead to a greater density of foraging individuals, reducing the distance between them during foraging. (Prediction 10).

To test these predictions, we studied how Little Swifts adjust their aerial foraging behavior to varying insect abundances in the airspace. Using radar and biotelemetry data, we reveal bird response to food abundance in relation to foraging distances, timing, foraging duration, and speed, as well as the frequency of colony visits and the distance between individuals. Our findings shed light on how aerial foragers may optimize their movement and behavior in response to highly dynamic environmental conditions.

## Results

The movement traffic rate (MoTR) (1207.7 ± 566.7 insects $km^{-1}$ $hr^{-1}$) varied substantially between different days during the swifts' breeding season, with a minimum of 164.4 and a maximum of 2518.9 insects $km^{-1}$ $hr^{-1}$ (n = 31 days; *Figure 1A*). No seasonal trend was found in MoTR (Spearman's rank correlation between the ordinal date and the MoTR, $\rho$ = −0.007, p=0.971, n = 31 days; *Table 1*). We found a significant negative effect of the MoTR on the swifts' average distance from the breeding colony (Prediction 1) (estimate <−0.001, $t$ = −5.27, p<0.001, n = 31 days, Gamma Generalized Linear Model [GLM]; *Figure 1B*). Similarly, a significant negative effect of MoTR was also found in relation to the birds' maximum distance from the breeding colony (Prediction 2) (estimate = −1.818, $t$ = −3.52, p=0.001, n = 31 days, Gaussian GLM; *Figure 1C*). We found no effect of MoTR on the daily route (Prediction 3) (estimate <−0.001, $t$ = −1.65, p=0.123, n = 15 days, Gamma GLM) and on the duration of foraging (Prediction 4) (estimate = 0.029, $t$ = 1.05, p=0.315, n = 15 days, Gaussian GLM). The frequency of visits at the breeding colony (Prediction 5) (see the average model in *Table 2*) was significantly and positively affected by MoTR (estimate = 0.001, $t$ = 3.78, p<0.001, n = 31 days, Gamma GLM; *Figure 1D*) and negatively affected by the distance of the birds from the breeding colony (estimate <−0.001, $t$ = 2.03, p=0.043, n = 31 days). We found that there was no effect of MoTR on the average flight speed (Prediction 6) (estimate <−0.001, $t$ = −1.33, p=0.193, n = 31 days, Gaussian GLM). The time of arrival at the breeding colony for nighttime roosting was significantly and negatively affected by the MoTR (Prediction 7) (estimate = −0.011, $t$ = −2.27, p=0.034, n = 23 days, Gaussian GLM), such that birds arrived earlier to roost in days characterized by abundant insect prey. The departure time from the breeding colony following overnight roosting has resulted in a consistently observed duration of nighttime roosting (10.45 ± 0.68 hr). This duration showed no correlation with the preceding day's MoTR (Prediction 8) (estimate = 0.002, $t$ = 0.26, p=0.801, n = 20, Gaussian GLM). Conversely, it was significantly and positively influenced by the evening arrival time to the colony on the prior day (Prediction 9a) (estimate = 0.634, $t$ = 2.81, p=0.016, n = 14 days, Gaussian GLM; *Figure 1E*). Furthermore, the departure time from the roost exhibited no association with MoTR of the same day (Prediction 9b) (estimate = −0.005, $t$ = −1.07, p=0.297, n = 20, Gaussian GLM). MoTR significantly and negatively affected (estimate <−0.001, $t$ = −3.12, p=0.004, n = 31 days, Gamma GLM) the distance between individuals (Prediction 10), while, as expected, the distance between individuals was significantly and positively correlated with the distance from the colony (estimate <0.001, $t$ = 5.02, p<0.001, n = 31 days; *Figure 1F*).

## Discussion

### Movement optimization during breeding

Our study provides novel insights regarding the optimal foraging of aerial insectivores by uniquely employing advanced tools to simultaneously track the movement and behavior of insectivore foragers and the dynamics of their insect prey aloft. We observed a reduction in average and maximum flight distance (Predictions 1 and 2) from the breeding colony in relation to MoTR, indicating that swifts can identify insect prey abundance and accordingly modify their flight distance and avoid using distant foraging locations when sufficient prey is found near the breeding colony. These results indicate

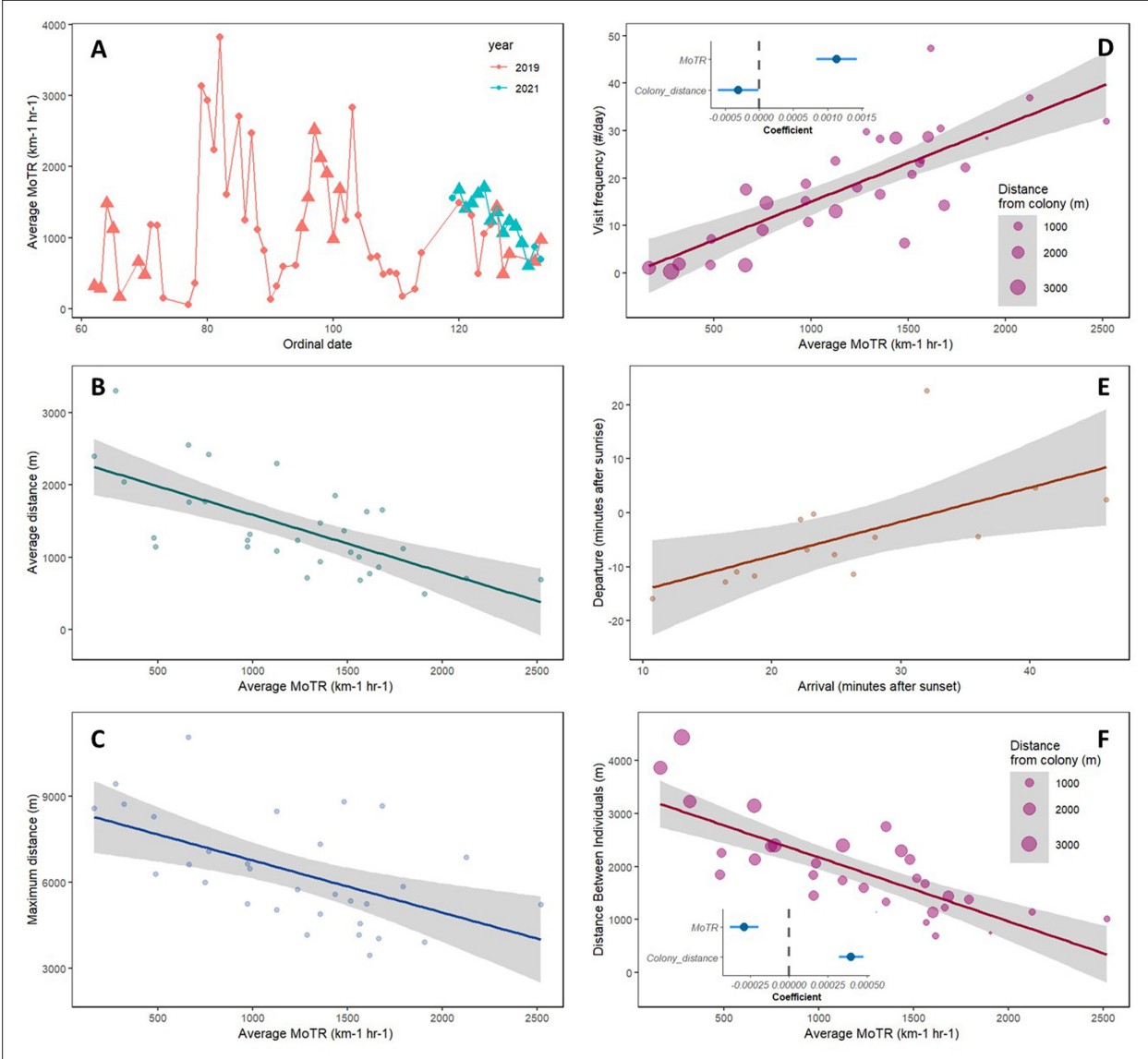

**Figure 1.** Insect traffic rate (movement traffic rate [MoTR]) and its effects on the aerial foraging of Little Swifts. (**A**) Average daily insect abundance in relation to an ordinal date. Triangles represent days when data allowed examining swift movement in relation to MoTR. MoTR varied across days within the swifts' breeding season by more than an order of magnitude. (**B**) The effect of MoTR on the average flight distance from the breeding colony. (**C**) The effect of MoTR on the maximal flight distance from the breeding colony. (**D**) The effect of MoTR on the average daily frequency of visits at the breeding colony; inset: coefficient value and confidence intervals of the coefficient resulting from the model testing the effects of MoTR and distance from the breeding colony on the frequency of visits. (**E**) The relationship between the time of departure from the breeding colony in the morning after the overnight stay and the time of arrival to the colony prior to the overnight stay the previous evening. (**F**) The effect of MoTR on the daily average distance between foraging individuals; inset: coefficient value and confidence intervals of the coefficient resulting from the model testing the effects of MoTR and distance from the colony on the distance between individuals.

The online version of this article includes the following figure supplement(s) for figure 1:

**Figure supplement 1.** An expected increase in the average distance between individuals with an increase in the distance from the breeding colony (black circle in the center of the figure).

that a significant decrease in insect abundance may lead swifts to expend more energy foraging in distant areas from the breeding colony, potentially impacting parental flight efficiency. Providing food to the young is a critical and enduring activity in bird life, influencing physiology (*Karell et al., 2009*; *Bukaciński et al., 1998*), immunity (*Appleby et al., 1999*), and survival (*Bukaciński et al., 1998*; *Brinkhof and Cavé, 1997*). Consequently, a reduction in flying insect abundance forcing birds to forage farther from the colony could have broad implications for the reproduction, survival, and

**Table 1.** Summary of the statistical analyses.

| Dependent variable | Independent variable | Estimate | t value | p value | Sample size (days) | Statistical test |
|---|---|---|---|---|---|---|
| Seasonal trend: average daily MoTR | Ordinal date | –0.007 | | 0.971 | 31 | Spearman's rank correlation |
| Prediction 1: Average distance from breeding colony | Average daily MoTR | <–0.001 | –5.27 | <0.001 | 31 | Gamma GLM |
| Prediction 2: Maximum distance from breeding colony | Average daily MoTR | –1.818 | –3.52 | 0.001 | 31 | Gaussian GLM |
| Prediction 3: Daily route | Average daily MoTR | <–0.001 | –1.65 | 0.123 | 15 | Gamma GLM |
| Prediction 4: Daily duration of foraging | Average daily MoTR | 0.029 | 1.05 | 0.315 | 15 | Gaussian GLM |
| Prediction 5: Frequency of visits at breeding colony | Average daily MoTR (the first of two independent variables) | 0.001 | 3.78 | <0.001 | 31 | Gamma GLM |
| | Distance from breeding colony (the second of two independent variables) | <–0.001 | 2.03 | 0.043 | 31 | |
| Prediction 6: Average flight speed | Average daily MoTR | <–0.001 | –1.33 | 0.193 | 31 | Gaussian GLM |
| Prediction 7: Evening arrival time to breeding colony | Average daily MoTR | –0.011 | –2.27 | 0.034 | 23 | Gaussian GLM |
| Prediction 8: Departure from breeding colony (overnight stay) | MoTR on the previous day | 0.002 | 0.26 | 0.801 | 20 | Gaussian GLM |
| Prediction 9a: Departure from breeding colony (overnight stay) | Evening arrival time to breeding colony (previous day) | 0.634 | 2.81 | 0.016 | 14 | Gaussian GLM |
| Prediction 9b: Departure from breeding colony (overnight stay) | MoTR on the same day | –0.005 | –1.07 | 0.297 | 20 | Gaussian GLM |
| Prediction 10: Distance between individuals | Average daily MoTR (the first of two independent variables) | <–0.001 | –3.12 | 0.004 | 31 | Gamma GLM |
| | Distance from breeding colony (the second of two independent variables) | <0.001 | 5.02 | <0.001 | 31 | |

GLM, Generalized Linear Model; MoTR, movement traffic rate.

population ecology of insectivores. Nevertheless, we investigated the impact of MoTR on the total daily route and foraging duration (Predictions 3 and 4). Our findings revealed no significant effects, suggesting that daily energy expenditure attributed to flight behavior does not exhibit a consistent pattern in response to the highly variable insect prey abundance and the associated shifts in swift flight behavior (higher proximity to the colony when prey is abundant).

While the theory of central-place foraging suggests that traveling to a distant destination is an expensive investment in terms of time and energy utilization compared to traveling to a nearby destination (*Orians and Pearson, 1979*; *Bell, 1990*; *Olsson et al., 2008*), our findings indicate that the birds may optimize their feeding rate (Prediction 5) to the young by staying close to the colony when food is abundant. We found that the frequency of colony visits was positively affected by MoTR (*Figure 1E*), indicating high provisioning rates when food was abundant, which supports an increase in the overall energy brought to the nestlings (*McCarty, 2002*). Thus, even when the birds foraged close to the colony under optimal conditions, the shorter traveling distance did not necessarily confer lower

**Table 2.** Top models (ΔAIC <2) for colony visit frequency.
The best model includes 'Distance from breeding colony' and 'Average daily MoTR'.

| Model | Intercept | Distance from breeding colony | Average daily MoTR | df | LogLik | AICc | Delta | Weight |
|---|---|---|---|---|---|---|---|---|
| 4 | 2.174 | <–0.001 | 0.001 | 4 | –108.2 | 225.9 | 0.00 | 0.626 |
| 3 | 1.093 | | 0.001 | 3 | –110.1 | 227.1 | 1.13 | 0.356 |

flight-related energetic expenditure because more return trips were made. Rather, it is the ability to provide more food to the young, by foraging close to the colony, that is being optimized, to benefit the reproductive output of the birds.

The availability of resources in a bird's habitat may affect the length of its daily route (*Stauss et al., 2005*), while others show no significant correlation (*Tremblay et al., 2005*). We found that the swifts maintained rather constant flight effort, regardless of the abundance of their prey. Similarly, foraging duration was also not related to MoTR. Further, our results suggest that food abundance had no significant impact on flight speed (Prediction 6). Consequently, our results support the idea that birds optimize food provisioning to the young during breeding, which could increase the birds' reproductive success at the expense of foraging energetics considerations. Another property of food provisioning to the young that may affect energy intake is the size of the load, but unfortunately, we have no information on whether the load size brought to the nest varied with insect abundance.

## Behavior optimization during breeding

Birds may adjust their foraging timing to optimize food intake (*Bednekoff and Houston, 1994*; *Bonter et al., 2013*; *Amichai and Kronfeld-Schor, 2019*; *Salamolard and Weimerskirch, 1993*). Our findings reveal that when insect prey was abundant in the airspace, the swifts' evening arrival time (Prediction 7) at the breeding colony was earlier than in days when insects were scarce. This result aligns with prior research on the predation risk-food availability trade-off, indicating that birds tend to avoid foraging during twilight hours due to elevated predation risk during this period (*Bonter et al., 2013*; *Lima, 1988*).

The availability of insects did not significantly influence the departure time (Prediction 8) from the colony after an overnight stay on both the same and previous days. Yet, the morning departure time was positively and significantly correlated with the time of arrival at the overnight roosting on the previous day. This result suggests a link between these specific behavioral features related to roosting timing. A possible explanation could be that birds arriving at the colony relatively early in the evening may be hungrier the following day, and this hunger may cause an earlier departure for foraging the following morning (Prediction 9a). Also, since these birds fed their young earlier, they may prefer to start foraging earlier the following morning, and thereby provide more food to their young in the morning to compensate for the early termination of feeding on the previous day (Prediction 9b). Further research is needed to establish the causes of this interesting relationship.

The influence of resource abundance on social foraging in aerial insectivorous birds remains a largely unexplored topic, despite its potential impact on bird fitness (*Giraldeau and Caraco, 2000*), energy intake (*Brown, 1988*; *Giraldeau and Caraco, 2000*; *Beauchamp, 1998*), predator avoidance (*Giraldeau and Caraco, 2000*; *Sorato et al., 2012*), and food acquisition dynamics (*Cvikel et al., 2015*; *Bijleveld et al., 2015*). Our findings suggest that when food is abundant the distance between foraging individuals (Prediction 10) is reduced, and this distance increases when food is scarce. A possible explanation for these findings is that when individuals forage at an increasing distance from the breeding colony (*Figure 2*) they may be too far from each other to detect each other and forage together in patchily distributed insect-rich patches in the airspace. When foraging closer to each other, local enhancement of individuals may take place when an effective foraging area is discovered (*Krebs, 1974*; *Harel et al., 2017*). Thus, swifts likely benefit from the advantages of local enhancement during periods of abundant food (*Brown, 1988*; *Flemming et al., 1992*; *Krebs, 1974*), but this enhancement might be limited when food is scarce.

## Central-place foraging

Many studies on central-place foraging examined foraging characteristics in relation to the distance and quality of the foraging patch (*Holder and Polis, 1987*; *Charnov, 1976*; *Bell, 1990*; *Olsson et al., 2008*; *Bryant and Turner, 1982*; *Kacelnik and Cuthill, 1990*; *Rosenberg and McKelvey, 1999*; *Elgin et al., 2020*). Our research deals with the abundance of food in the aerial habitat, which is highly dynamic, as corroborated by our findings that insect abundance varied greatly, by more than an order of magnitude, between different days during the swifts' breeding period. Although insect abundance aloft varies with time, it is not clear to what extent it varies in space as several studies suggested that insect bioflow is correlated over large spatial scales (*Chapman et al., 2002*; *Welti et al., 2022*; *Goulson et al., 2005*). Hence, patches of high insect concentration might be transient and spatially

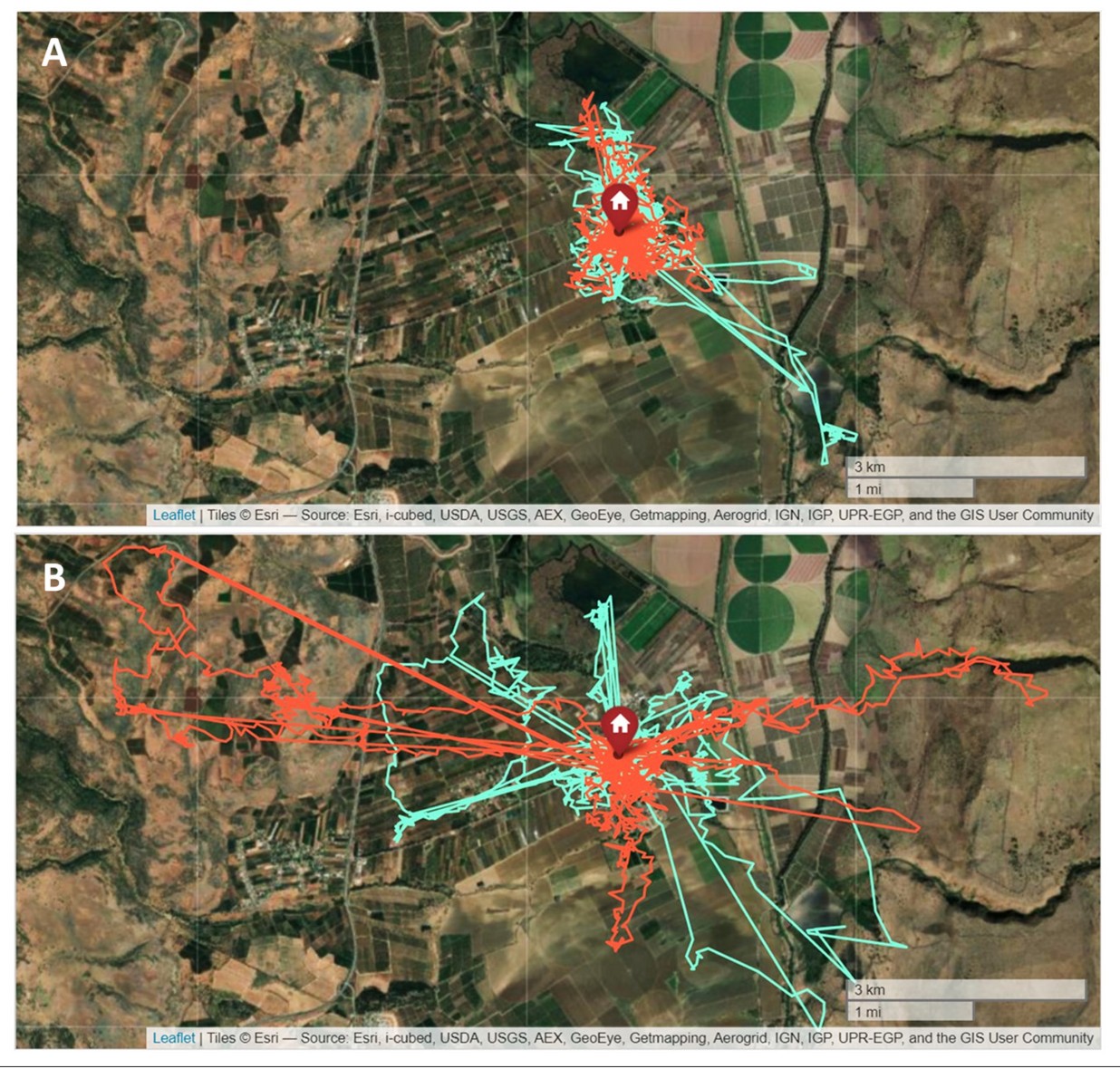

**Figure 2.** Foraging range in relation to insect abundance. Differences in the movement routes of two individuals (marked in light blue and orange) on two consecutive days that were characterized by large differences in movement traffic rate (MoTR). (**A**) 09.04.2019 (average MoTR = 1904.2 insects km$^{-1}$ hr$^{-1}$). (**B**) 10.04.2019 (average MoTR = 983.5 insects km$^{-1}$ hr$^{-1}$).

variable; thus, further study is needed to characterize the spatial properties of insect bioflow. It is known that insect concentrations occur under specific meteorological conditions, for example, on the edges of air fronts (*Reynolds et al., 2018*), as well as near topographic features where the wind may subside (*Drake and Reynolds, 2012*). We call for a better description of the spatial properties of insects in the aerial habitat, specifically the horizontal and vertical distribution of insects in the airspace and how it might be affected by different factors, including topography, coastlines, and weather conditions. Our study, with its primary focus elsewhere, did not delve into this aspect. Nonetheless, the availability of today's advanced technological tools attests to the feasibility of conducting such research.

## Integrating advanced tracking systems for ecological research

Due to its nature, aeroecological research is limited by the paucity of appropriate tools to track aerial animals and their dynamic environment in detail (*Nathan, 2005*; *Kunz et al., 2008*). Several recent

technological developments facilitated a better grasp of the aerial environment, allowing the examination of various aspects of aerial ecology that were impossible to test in the past or that were explored only with coarse resolution (*Nathan et al., 2005*). The combination of two advanced systems, namely ATLAS and the BirdScan-MR1 radar, allows, for the first time, a detailed investigation of fundamental aspects of animal foraging in the airspace through the study of predator–prey interactions between Little Swifts and their insect prey. Recent progress in wildlife tracking technologies enables new insights into the movement patterns of animals, including their causes, consequences, and underlying mechanisms, facilitated by the integration of complementary tools (*Nathan et al., 2022*), as demonstrated here. Specifically, the unique combination of advanced technologies to expand the boundaries of aeroecological research can be expanded and further utilized for understanding how changes in the aerial habitat that are related to human activities may affect organisms that live in this unique and dynamic habitat (*Hallmann et al., 2014*; *Møller, 2019*). These insights may play a crucial role in the conservation of aerial insectivores that are dramatically affected by human related alteration, including habitat degradation and the use of pesticides (*Stanton et al., 2016*; *Nocera et al., 2012*).

## Methods

Little Swifts breed in Israel between March and September, during which they complete two breeding cycles. In the swifts' breeding colony, dozens of pairs inhabit interconnected nests crafted from feathers and stems. Each nest features a narrow entrance designed to deter predators and obscure visibility inside. This setup presents challenges in accurately assessing nesting conditions and determining the age of chicks within the nests. Both partners incubate alternately, and during the night, they both stay in the nest. The incubation period lasts 18–22 days, and fledging occurs 35–40 days after hatching. Both parents participate in the feeding of the young (*Shirihai et al., 1996*; *Paz, 1987*).

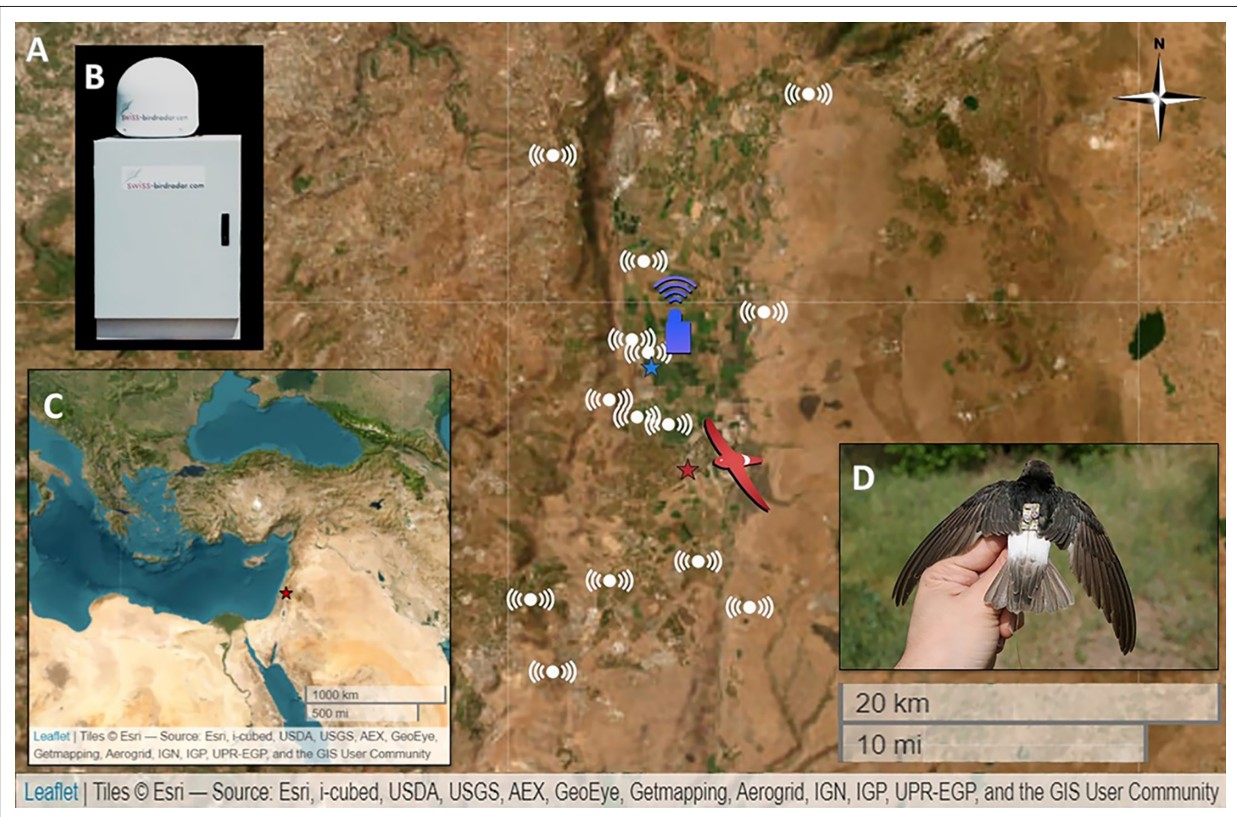

**Figure 3.** The research system. (**A**) Map of the Hula Valley, Israel, and the surrounding area. The red star represents the location of the Little Swifts' breeding colony. The blue star depicts the location of the radar. White markings indicate the locations of the Advanced Tracking and Localization of Animals in Real-Life Systems' (ATLAS) antennas. (**B**) The BirdScan-MR1 radar. (**C**) The location of the research system in northeastern Israel within the Middle East is indicated by a red star. (**D**) A Little Swift with an ATLAS tag.

We studied swifts in a breeding colony located in the center of the Hula Valley in northeastern Israel (33.05°N/35.59°E). The valley consists of a mosaic of agricultural land with various crops, mainly deciduous tree plantations and open field crops, as well as wetlands and urbanized areas. Our field observations suggest that there are about 30–40 nesting pairs in the colony.

## Swift movement data collection

During March–May of 2019 and 2021, employing a 9 m mist net outside the breeding colony, we captured Little Swifts during their early morning departure after the night stay. Our bird-trapping activities were conducted under permits (2019-42174 and 2021-42762) of the Israel Nature and Parks Authority. Captured swifts were measured and ringed with a standard aluminum ring to allow individual identification. We equipped 32 swifts with ATLAS transmitters weighing 1–1.15 g, less than 5% of the body mass of each individual.

The ATLAS is a reverse GPS-like system that operates using time-difference-of-arrival of radio waves to base stations (antennas), recording the horizontal locations of tagged animals within the system's coverage area at high frequency (the tags transmitted every 8 s) and spatial accuracy (~10 m). The system includes antennas deployed throughout the Hula Valley and the surrounding area (*Figure 3*), facilitating the calculation of the spatial position of the radio transmitters that emit a unique ID signal for each transmitter. The transmitters were affixed to the swifts using a backpack harness positioned between the back feathers, secured with Perma-Type Surgical Cement (Perma-Type Company Inc, Plainville, CT). This adhesive naturally dries and allows the harness to fall off after several weeks (*Johnson et al., 1991*). Except for one tag that stopped transmitting immediately after release, the tags operated for periods of 0.3–39.8 days (X = 13.4 ± 10.4 days).

We analyzed a total of 841,342 localizations during days in which we obtained both bird movement data from the ATLAS and insect abundance data from the radar (see below). The data were collected over a total of 31 days (19 days in 2019 and 12 days in 2021). Because swifts are active during the daytime, we used only ATLAS data from the main activity hours of the swifts during the day, from sunrise to sunset (*Thieurmel and Elmarhraoui, 2019*) (personal observations and movement data obtained from the ATLAS).

We applied several filters to reduce inaccuracies in the movement tracks as a result of localization errors (*Gupte et al., 2022*). Since there is no accurate information about the maximum flight speed of Little Swifts, we relied on the maximum flight speed of the Common Swift (*Henningsson et al., 2010*) to filter out tracks with a flight speed that exceeded 30 m/s (9.6% of the raw data). We additionally utilized the standard error of the localization (StdLoc) to assess position quality, identifying outliers (1.5 times the interquartile range) of StdLoc (*Tukey, 1977*). Setting an upper limit at 30.1 m, we filtered out positions with low accuracy, amounting to 10.7% of the data. Additionally, we applied a minimum threshold of 4 (*Arnon et al., 2023*) for the Number of ATLAS Base Stations (NBS) receiving a tag's signals during each transmission to filter out localizations with low confidence of accuracy (4.0% of the data; range of NBS after filtering: 4–14, $\bar{X} = 6.6 \pm 1.9$ NBS). We then excluded tracks in which consecutive locations were more than 500 m away from each other (0.7% of the data), likely representing an error in the automatic calculation of the tag's position. The filtering process removed a total of 24.5% of the raw data. To ensure the overall dataset represented the movement of all birds without being influenced by the unusual behavior of a few, we excluded data from days with fewer than four active tags (range of number of tags after filtering: 4–10, mean ± SD = 6.9 ± 1.5 tags per day). This threshold eliminated days with a small number of tagged birds recorded (24.8% of the data). As a result, a total of 49.4% of the original raw data was excluded to maintain a high level of reliability and accuracy; analysis was based on 415,420 positions, with a mean of 1491 ± 899 locations per tag per day.

## Movement analysis

To examine bird movement (*Figure 2*) and behavior, we calculated the average and maximum distance from the breeding colony. To determine the average daily route and duration of foraging, we analyzed data from 15 days with a minimum of 10 hr of consistent tag activity, excluding cases of tag malfunction or battery issues. There was no tag reception when the swifts entered the building that housed their breeding colony, allowing easy determination of when they visited the colony. To standardize the effect of day length on the foraging duration, we calculated and subtracted the day length from the total daily foraging time (Day duration – Daily foraging duration = Net foraging duration). The

resulting data represent the foraging duration in relation to sunrise and sunset, independent of day length. To characterize the rate of visitation to the breeding colony, we defined visits as events in which birds stayed in the colony for at least 60 s. The time of arrival to the breeding colony for night roosting was calculated as minutes after sunset, within a 60 min window around sunset, and the same was done for the morning departure time, but in relation to sunrise. We calculated the average departure and arrival time of all active tags for each day.

We omitted days when the nighttime arrival to the colony was missing (e.g., days when the battery ran out during the day) or days when the morning departure time from the colony was missing. Consequently, we were left with 23 days of arrival data, 20 days of departure data, and 20 days of departure in which data existed regarding the abundance of insects (below) on the previous day. To compute the average distance between individuals, we calculated the average position every 5 min for each bird and omitted cases where we had simultaneous location data of less than four individuals. We then calculated the daily average of the distance between individuals.

## Radar measurements of insect abundance

Studies have shown that environmental variables like temperature and wind significantly influence the spatial abundance of insects across different crop areas (*Goulson et al., 2005*; *Gruebler et al., 2008*). To estimate the abundance of insects aloft, we used the daily average MoTR of aerial insects recorded by the BirdScan-MR1 radar (*Knop et al., 2023*) (Swiss-birdradar, Winterthur, Switzerland) located in the Hula Valley (33.06°N/35.35°E), 6.5 km north of the Little Swifts' breeding colony. The radar is capable of detecting flying animals, including songbirds, waterbirds, bird flocks, large single birds, and insects, by classifying them according to the patterns of the echo (*Zaugg et al., 2008*; *Zaugg et al., 2017*). The Radar Cross Section (RCS) quantifies the reflectivity of a target, serving as a proxy for size by representing the cross-sectional area of a sphere with identical reflectivity to water, whose diameter equals the target's body length (*Chilson et al., 2018*). Recent findings indicate that the BirdScan-MR1 radar can detect insects with an RCS as low as 3 mm² (*Haest et al., 2024*), with decreasing detection probability at increasing altitudes. The detection threshold, defined by the STC setting, was 93 dBm, and the transmit power was 25 kW (*Haest et al., 2024*). These capabilities make the radar suitable for locating the primary prey of swifts, which typically range in size from 1 to 16 mm (*Collins et al., 2010*). In addition, the radar automatically calculates the height, speed, and direction of movement of the object. The radar has an upward-pointing antenna that picks up objects passing within a 90–120° vertical cone over it. Insects are recorded by the radar from a height of about 50 m above ground level up to a height of about 700 m above the ground. We calculated the daily averaged MoTR from 5 am to 8 pm local time as a standard measure of insect abundance rates. This was done by counting insects per hour across a 1 km cross-section and averaging these counts over a single day, allowing for comparisons of aerial movement between different days (*Liechti et al., 2019*). We matched the insect data obtained from the radar with the swift movement data obtained from the ATLAS.

## Statistical analysis

Using the 'stats' package in R (*R Development Core Team, 2021*), we applied GLMs and Spearman's correlations to explore the effects of the MoTR (continuous independent variable) on the movement and behavior parameters of the swifts during the breeding season. If the GLM, with more than one explanatory variable, had a ΔAIC <2 relative to other models, we employed the MuMIn (*Barton, 2021*) package to generate an average model. Specifically, we investigated how the distance between individuals is influenced by both the distance of birds from the colony and MoTR. Accounting for the expected increase in individual distance when flying farther from the breeding colony due to a larger air volume occupied by the moving birds, these factors were integrated into our GLM analysis. The same approach was applied in modeling the frequency of visits to the colony. To distinguish the effects of breeding colony distance and insect abundance on the distance between individuals, our GLM incorporated both variables, ensuring a comprehensive understanding of the impact of distance from the colony (*Figure 1—figure supplement 1*). In the model testing which factors affected the time of arrival at the colony, the frequency of visits was highly correlated with MoTR and was therefore removed from the model at an initial stage. The departure time from the colony and the length of the daily route did not significantly affect the arrival time and were left out of the model at a later

stage. Consequently, the final model included only MoTR as an explanatory factor for colony arrival time. We additionally tested if the time of departure from the breeding colony after the overnight stay was related to three explanatory variables, MoTR, MoTR on the previous day, and the arrival time to the colony for the overnight stay on the previous day. We used the fitdistrplus package (*Delignette-Muller and fitdistrplus, 2015*) to identify the appropriate distribution for each GLM. We used R version 4.1.2, *R Development Core Team, 2021* for all the statistical analyses. Data reported are average ± SD unless noted otherwise, and the analyses were two-tailed with a critical $\alpha$ = 0.05.

## Acknowledgements

We thank Yosef Kiat, Eve Miller, Ayla Rimon, Stav Shay, and Gev Galili for their help with the fieldwork and Yoni Vortman, Yoav Bartan, Yotam Orchan, and Anat Levi for logistical support. The study was supported by a grant from the KKL-JNF (Kanfei KKL, contract no. 60-05-675-18) and the Israel Science Foundation grants ISF-2333/17 and ISF-1653/22 (to NS), ISF-965/15 (to RN and ST), and ISF-1919/19 (to ST). The ATLAS work was funded by the Minerva Foundation, the Minerva Center for Movement Ecology, the Adelina and Massimo Della Pergola Professor of Life Sciences to RN.

## Additional information

### Funding

| Funder | Grant reference number | Author |
|---|---|---|
| KKL-JNF | 60-05-675-18 | Nir Sapir |
| Israel Science Foundation | ISF-2333/17 and ISF-1653/22 | Nir Sapir |
| Israel Science Foundation | ISF-965/15 | Sivan Toledo Ran Nathan |
| Israel Science Foundation | ISF-1919/19 | Sivan Toledo |
| Minerva Center for Movement Ecology, Hebrew University of Jerusalem | | Ran Nathan |

The funders had no role in study design, data collection and interpretation, or the decision to submit the work for publication.

### Author contributions
Itai Bloch, Conceptualization, Formal analysis, Investigation, Visualization, Methodology, Writing – original draft; David Troupin, Formal analysis, Visualization; Sivan Toledo, Resources, Funding acquisition; Ran Nathan, Resources, Writing - review and editing; Nir Sapir, Conceptualization, Resources, Formal analysis, Supervision, Funding acquisition, Investigation, Methodology, Writing - review and editing

### Author ORCIDs
Itai Bloch https://orcid.org/0000-0002-7514-2121
Nir Sapir https://orcid.org/0000-0002-2477-0515

Reviewer #1 (Public review): https://doi.org/10.7554/eLife.96573.4.sa1
Reviewer #2 (Public review): https://doi.org/10.7554/eLife.96573.4.sa2
Author response https://doi.org/10.7554/eLife.96573.4.sa3

## Additional files

### Supplementary files
MDAR checklist

## Data availability

The dataset is available via Dryad.

The following dataset was generated:

| Author(s) | Year | Dataset title | Dataset URL | Database and Identifier |
|---|---|---|---|---|
| Troupin D, Toledo S, Nathan R, Sapir N | 2025 | Combining radio-telemetry and radar measurements to test optimal foraging in an aerial insectivore bird | https://doi.org/10.5061/dryad.xwdbrv1qh | Dryad Digital Repository, 10.5061/dryad.xwdbrv1qh |

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
