## [Editor Report · eLife Assessment]

This **important** study enhances our understanding of the foraging behavior of aerial insectivorous birds. Using **solid** methodology, the authors have collected extensive data on bird movements and prey availability, which in turn provide support for the main claim of the study. The work will be of broad interest to behavioral ecologists.

---

## [Referee Report · Reviewer #1 (Public review)]

This study tests whether Little Swifts exhibit optimal foraging, which the data seem to indicate is the case. This is unsurprising as most animals would be expected to optimize the energy income : expenditure ratio, however it hasn't been explicitly quantified before the way it was in this manuscript.

The major strength of this work is the sheer volume of tracking data and the accuracy of those data. The ATLAS tracking system really enhanced this study and allowed for pinpoint monitoring of the tracked birds. These data could be used to ask and answer many questions beyond just the one tested here.

The major weakness of this work lies in the sampling of insect prey abundance at a single point on the landscape, 6.5 km from the colony. This sampling then requires the authors to work under the assumption that prey abundance is simultaneously even across the study region. It may be fair to say that prey populations might be correlated over space but are not equal. It is uncertain whether other aspects of the prey data are problematic. For example, the radar only samples insects at 50m or higher from the ground - how often do Little Swifts forage under 50m high?

The finding that Little Swifts forage optimally is indeed supported by the data, notwithstanding some of the shortcomings in the prey abundance data. The authors achieved their aims and the results support their conclusions.

At its centre, this work adds to our understanding of Little Swift foraging and extends to a greater understanding of aerial insectivores in general. While unsurprising that Little Swifts act as optimal foragers, it is good to have quantified this and show that the population declines observed in so many aerial insectivores are not necessarily a function of inflexible foraging habits. Further, the methods used in this research have great potential for other work. For example, the ATLAS system poses some real advantages and an exciting challenge to existing systems, like MOTUS. The radar that was used to quantify prey abundance also presents exciting possibilities if multiple units could be deployed to get a more spatially-explicit view.

To improve the context of this work, it is worth noting that this research goes into much further depth than any previous studies on a similar topic in several flycatcher and swallow species. A further justification is posited that this research is needed due to dramatic insect population declines, however, the magnitude and extent of such declines are fiercely debated in the literature.

---

## [Referee Report · Reviewer #2 (Public review)]

Summary:

Bloch et al. studied the relationships between aerial foragers (lesser swifts) tracked using an automated radio telemetry system (Atlas) and their prey (flying insects) monitored using a small vertical-looking radar (BirdScan MR1). The aim of the study was to check whether swifts optimise their foraging according to the abundance of their prey. The results provide evidence that small swifts can increase their foraging rate when aerial insect abundance is high, but found no correlation between insect abundance and flight energy expenditure.

Key points:

This study fills gaps in fundamental knowledge of prey-predator dynamics in the air. It describes the coincidence between the abundance of flying insects and the characteristics derived from monitoring individual swifts.

Weaknesses:

The paper uses assumptions largely derived from optimal foraging theory, but mixes up the form of natural selection: parental energy, parental survival (predation risk), nestling foraging and reproductive success. The results are partly inconsistent, and confounding factors (e.g., the brooding phase versus the nestling phase) remained ignored. In conclusion, the analyses performed are insufficient to rigorously assess whether lesser swifts are optimising their foraging beyond making shorter foraging trips.

The filters applied to the monitoring data are necessary but may strongly influence the characteristics derived based on maximum or mean values. Sensitivity tests or the use of characteristics that are less dependent on extreme values could provide more robust results.

---

## [Author Response]

The following is the authors’ response to the previous reviews.

**Reviewer #1 (Recommendations for the authors):**
I am generally satisfied with the authors' revisions and response to my previous comments. I have amended my previous review.

We thank Reviewer #1 for his valuable comments and suggestions, which improved this manuscript.

Thank you for considering the comments in your revised version. I still feel a strong mismatch between the claims of optimal foraging behaviour and the results with little compelling evidence.On terminology: MTR means Migration Traffic Rates. The authors responded that in their study, MTR is defined as Movement traffic rates. I have two problems with this definition: (i) it creates confusion in the literature on the definition of MTR, (ii) a traffic inherently describes a movement, and this pleonasm is not necessary.

We revised the acronyms in this article, replacing MTR with MoTR to clearly distinguish between Migration Traffic Rate (MTR) and Movement Traffic Rate (MoTR).

Minimal size of insects: Please detail radar settings (power sent, STC; detection thresholds). These parameters define the minimal size of the detected animals.

We added the following paragraph to provide additional information regarding the radar's detection capabilities:

" with decreasing detection probability at increasing altitudes. The detection threshold, defined by the STC setting, was 93 dBm, and the transmit power was 25 kW."